# Myosin Binding Protein-C Forms Amyloid-Like Aggregates In Vitro

**DOI:** 10.3390/ijms22020731

**Published:** 2021-01-13

**Authors:** Liya G. Bobyleva, Sergey A. Shumeyko, Elmira I. Yakupova, Alexey K. Surin, Oxana V. Galzitskaya, Hiroshi Kihara, Alexander A. Timchenko, Maria A. Timchenko, Nikita V. Penkov, Alexey D. Nikulin, Mariya Yu. Suvorina, Nikolay V. Molochkov, Mikhail Yu. Lobanov, Roman S. Fadeev, Ivan M. Vikhlyantsev, Alexander G. Bobylev

**Affiliations:** 1Laboratory of the Structure and Functions of Muscle Proteins, Institute of Theoretical and Experimental Biophysics, Russian Academy of Sciences, 142290 Pushchino, Russia; liamar@rambler.ru (L.G.B.); shumik92@gmail.com (S.A.S.); yakupova.mira@mail.ru (E.I.Y.); ogalzit@vega.protres.ru (O.V.G.); 2Laboratory of Bioinformatics and Proteomics, Institute of Protein Research, Russian Academy of Sciences, 142290 Pushchino, Russia; alan@vega.protres.ru (A.K.S.); marrruko@yandex.ru (M.Y.S.); m.u.lobanov@mail.ru (M.Y.L.); 3Biological Testing Laboratory, Branch of the Shemyakin–Ovchinnikov Institute of Bioorganic Chemistry, Russian Academy of Sciences, 142290 Pushchino, Russia; 4Laboratory of the Biochemistry of Pathogenic Microorganisms, State Research Centre for Applied Microbiology and Biotechnology, Obolensk, 142279 Serpukhov District, Russia; 5Department of Early Childhood Education, Himeji-Hinomoto College, 890 Koro, Kodera-cho, Himeji 679-2151, Japan; kiharah1234@gmail.com; 6Group of Experimental Research and Engineering of Oligomeric Structures, Institute of Protein Research, Russian Academy of Sciences, 142290 Pushchino, Russia; atim@vega.protres.ru; 7Laboratory of Applied Enzymology, FRC PSCBR, Russian Academy of Sciences, 142290 Pushchino, Russia; maria_timchenko@mail.ru; 8Laboratory of the Methods of Optical Spectral Analysis, Institute of Cell Biophysics, Russian Academy of Sciences, FRC PSCBR RAS, 142290 Pushchino, Russia; nvpenkov@rambler.ru; 9Laboratory for Structural Studies of the Translational Apparatus, Institute of Protein Research, Russian Academy of Sciences, 142290 Pushchino, Russia; nikulin@vega.protres.ru; 10Laboratory of NMR Investigations of Biosystems, Institute of Theoretical and Experimental Biophysics, Russian Academy of Sciences, 142290 Pushchino, Russia; nmolochkov@gmail.com; 11Laboratory of Pharmacological Regulation of Cell Resistance, Institute of Theoretical and Experimental Biophysics, Russian Academy of Sciences, 142290 Pushchino, Russia; fadeevrs@gmail.com

**Keywords:** muscle proteins, MyBP-C, amyloid, amyloid-like aggregation, protein aggregation, structural analysis, SAXS, CD, X-ray diffraction, DLS

## Abstract

This work investigated in vitro aggregation and amyloid properties of skeletal myosin binding protein-C (sMyBP-C) interacting in vivo with proteins of thick and thin filaments in the sarcomeric A-disc. Dynamic light scattering (DLS) and transmission electron microscopy (TEM) found a rapid (5–10 min) formation of large (>2 μm) aggregates. sMyBP-C oligomers formed both at the initial 5–10 min and after 16 h of aggregation. Small angle X-ray scattering (SAXS) and DLS revealed sMyBP-C oligomers to consist of 7–10 monomers. TEM and atomic force microscopy (AFM) showed sMyBP-C to form amorphous aggregates (and, to a lesser degree, fibrillar structures) exhibiting no toxicity on cell culture. X-ray diffraction of sMyBP-C aggregates registered reflections attributed to a cross-β quaternary structure. Circular dichroism (CD) showed the formation of the amyloid-like structure to occur without changes in the sMyBP-C secondary structure. The obtained results indicating a high in vitro aggregability of sMyBP-C are, apparently, a consequence of structural features of the domain organization of proteins of this family. Formation of pathological amyloid or amyloid-like sMyBP-C aggregates in vivo is little probable due to amino-acid sequence low identity (<26%), alternating ordered/disordered regions in the protein molecule, and S–S bonds providing for general stability.

## 1. Introduction

Protein aggregation is a rather widespread process in cells of a living organism. Amyloid aggregation is of great interest for researchers throughout the world due to the wide occurrence of amyloidoses in man and animals. These disorders are characterized by amyloid deposits, comprising misfolded proteins, in various human and animal organs and tissues. To date, more than 30 such proteins/peptides have been identified; they include, e.g., the Aβ peptide, the immunoglobulin heavy chain, β2-microglobulin [1]. These conditions include Alzheimer’s disease, Parkinson’s disease, type 2 diabetes mellitus, prion diseases, systemic amyloidoses [2,3,4,5]. Increasingly more attention has been paid to yet another type of amyloid aggregates—functional amyloids, whose formation is not related to disorders but is required to perform certain functions. Functional amyloids form such proteins as curli (*E. coli*) [6], tafi (*Salmonella* spp.) [7], chaplins (*Streptomyces coelicolor*) [8]. Functional amyloids also include spidroin fibrils occurring in spider silk threads [9], as well as Pmel17 fibrils in mammals [10]. 

All amyloid aggregates are known to have a cross-β quaternary structure [11]. However, it is not yet clear which mechanisms regulate the formation of pathological and functional amyloids in the organism. Research into the amyloid properties of multidomain muscle proteins, such as titin and myosin binding protein-C (MyBP-C), which consist of β-folded domains [12,13,14], can, in our mind, contribute to the insight into this problem. 

The family of myosin binding protein-C is a group of sarcomeric proteins from striated muscles. There are three paralogs of this protein that are coded for by separate genes. The paralog of slow skeletal MyBP-C (ssMyBPC) is expressed in skeletal muscle and is coded for by the *MYBPC1* gene. The paralog of fast skeletal MyBP-C (fsMyBP-C) is also expressed in skeletal muscle and is coded for by the gene *MYBPC2*. The paralog of cardiac MyBP-C (cMyBP-C) is expressed correspondingly in cardiac muscle MyBP-C (cMyBP-C) and is coded for by the gene *MYBPC3* [12]. These three paralogs consist of immunoglobulin-like (Ig) and fibronectin III-like (FnIII) domains (Figure 1) [12,13]. It has been found that *MYBPC1* possesses a high level of splicing, which its transcript may undergo [15]. In humans and mice, about 14 transcripts coding for the protein have been described. ssMyBP-C isoforms in humans and mice have molecular weights from 126 up to 132 kDa [15,16]. 

It is known that in sarcomeres of striated muscles MyBP-C binds to F actin [17,18,19,20], titin [21,22,23], obscurine [24], and myosin [20,25,26,27,28,29]. Recently, the interaction of MyBP-C with titin has been shown to be responsible for the C-protein’s localization in the C-zone of the sarcomere [23]. Binding of skeletal MyBP-C (sMyBP-C) to four and a half LIM protein 1 (FHL1) has been described [30]; this protein is known to be a critical regulator of striated muscle development [31]. Binding of ssMyBP-C to muscle-type creatine kinase (MM-CK) [32], and of cMyBP-C to calmodulin [33] and formin homology-2 domain-containing protein FHOD3 [34] has been shown. It is discussed that MyBP-C is an important regulator of striated muscle physiology [12].

This work investigated in vitro aggregation features and amyloid-like properties of rabbit skeletal myosin binding protein-C. We believe that, due to its ability to interact in vivo with various proteins [17,18,19,20,21,22,23,24,25,26,27,28,29], this protein is apt to aggregation. Our interest in MyBP-C aggregation is due to the results of our earlier research into the aggregation properties of titin, which also consists of immunoglobulin-like (Ig) and fibronectin III-like (FnIII) domains. Studies of smooth-muscle titin reveal the ability of this protein to form amyloid aggregates of a cross-β quaternary structure in vitro [35]. Although the molecular weight of titin exceeds that of MyBP-C more than tenfold, the presence of Ig and FnIII domains in molecules of these proteins suggests the same structural changes in aggregation. 

Using a large set of methods, such as DLS, TEM, AFM, CD, X-ray diffraction, and SAXS, this work investigated in vitro features of amyloid-like aggregation of rabbit sMyBP-C. 

## 2. Results

### 2.1. SDS-PAGE and HPLC–MS Analysis of sMyBP-C 

Figure 2 presents SDS-PAGE of the purified preparation of sMyBP-C. Two protein lanes are seen, which are, most probably, either two paralogs of sMyBP-C (fsMyBP-C and ssMyBP-C) or one of the paralogs of this protein and its degradation product (Appendix A). By the densitometry data, the molecular weight of the upper band of the protein is ~134 kDa; it is ~82% of the total amount (Appendix A). The molecular weight of the protein in the lower band is ~125 kDa; its amount, ~18%. HPLC–MS analysis confirmed the produced protein preparations to be the fsMyBP-C paralog (85% of sequence overlap) (Appendix A). 

### 2.2. Dynamic Light Scattering of sMyBP-C Aggregates 

The next objective of our study was to elucidate the aggregation rate of sMyBP-C. To clarify the time course of aggregate formation, we used DLS as a less invasive way to detect and characterize aggregates of different size [36,37]. Figure 3a–c shows a change in the autocorrelation function of scattered light upon formation of sMyBP-C aggregates at pH 7.0 over 1040 min. During the first 5-min incubation, the correlation function *g*_1_(*t*) was observed to decay in an almost mono-exponential way (Figure 3a), with the subsequent emergence of a pronounced shoulder at high correlation times. This is indicative of the formation of large aggregates with smaller diffusion coefficients. The dimensional peak of the aggregates formed in the first minutes of incubation, corresponding to this correlation time, cannot be recorded due to too large sizes not included in the size range of the method. 

After 20 min, the shoulder vanished, indicating a decrease in the aggregation process. At high correlation times, another shoulder emerged after 40 min and was observed after 60 and 180 min. This shoulder vanished by the 200th minute. Thus, we observed the second wave of sMyBP-C aggregation, which lasted within the interval of 20–200 min. Figure 3d–f shows the DLS data presented in coordinates of particle size distribution by volume/*R*_h_ (*R*_h_, the hydrodynamic radius). Prior to the formation of aggregates, only one well-resolved peak of an average *R*_h_ = ~5.6 nm (the dominating peak of ~99.9%) was observed. This peak corresponds, most likely, to sMyBP-C molecules (Figure 3d). We also observed one predominant dimensional fraction (the dominating peak of ~99.0%) within the entire measured period of time (Figure 3e,f). Over a period of 20–980 min, this dimensional peak was found to shift towards larger sizes up to *R*_h_ = ~13 nm. After 980 min, the size of this fraction ceased to increase. We assume that an increase of this dimensional fraction in size (from *R*_h_ = ~5.6 nm up to *R*_h_ = ~13 nm) is due to the formation of sMyBP-C oligomers. If the sizes of these oligomers in three dimensions are close to one another, then the registered protein particles (*R*_h_ = ~13 nm) may contain from 7 up to 10 monomers (*R*_h_ = ~5.6 nm) depending on the packing density. 

### 2.3. Electron Microscopy of sMyBP-C Aggregates 

Figure 4 shows electron micrographs of negatively stained sMyBP-C aggregates. After a 10-min aggregation (Figure 4a,d) and a 24-h aggregation (Figure 4b–f) at 4 °C in 0.15 M glycine-KOH, pH 7.0, sMyBP-C formed amorphous aggregates of more than 2 μm in size and spherical aggregates (supposedly oligomers) of ~40 nm in diameter. Thread-like sMyBP-C fibrils about 20 nm wide (Figure 4c,f) occurred less frequently. 

### 2.4. Atomic force Microscopy of sMyBP-C Aggregates 

AFM of sMyBP-C incubated for 24 h in a solution containing 0.15 M glycine-KOH, pH 7.0, revealed aggregates of different morphology and size. Large amorphous structures up to 4 µm long and ~7–15 nm high were observed (Figure 5a,b). Large extended structures ~100 nm high and ~100 nm wide also occurred (Figure 5c,d). Short extended and spherical aggregates were also seen (Figure 5b–d). Small spherical structures ~50 nm in diameter and ~10 nm high were, apparently, protein oligomers. Short extended structures up to 200 nm long, ~50–60 nm wide, and 4–6 nm high were also found (Figure 5b, Appendix A). Possibly, these structures are short protofibrils or a group of oligomers at a close distance one from another. The latter assumption is the most preferable as DLS revealed no particles of size corresponding to that of protofibrils. Interestingly, extended aggregates and spherical small structures were of the same height, and in most cases, the diameter of these spherical aggregates coincided with the width of extended structures (Figure 5b). 

### 2.5. X-ray Diffraction of sMyBP-C Aggregates 

X-ray diffraction analysis revealed ring-shaped diffuse X-ray reflections at 10 Å and 4.6 Å for sMyBP-C aggregates (Figure 6a, Appendix A). Similar reflections are seen in amyloids of a cross-β sheet quaternary structure [10,38,39,40]. The ~4.6 Å reflection is assumed to arise from the periodicity of hydrogen-bonded β-strands oriented perpendicular to the fiber axis, and the diffraction in the region of ∼10 Å is presumed to be related to the stacking of these sheets parallel to the fiber axis [40,41,42]. Proceeding from these data, sMyBP-C aggregates can be concluded to have a cross-β quaternary structure and, therefore, be amyloid. Two other reflections at 2.2 and 3.1 Å arise, most probably, from impurities in the sample. 

### 2.6. Circular Dichroism Analysis of sMyBP-C Secondary Structure

Figure 6b shows CD spectra of sMyBP-C before and after the formation of aggregates. No changes were detected in the secondary structure upon formation of the aggregates: the preparation after chromatography had a 4.5% α-helix and 30.7% β-structure, while the helix and β-structure contents in aggregated sMyBP-C were, respectively, 4.9% and 29.7%. Thus, amyloid aggregates are formed without changes in the secondary structure of sMyBP-C. 

### 2.7. Association of sMyBP-C Aggregates with Thioflavin T

ThT was used to confirm the amyloid nature of sMyBP-C aggregates (Figure 6c). The fluorescence intensity of ThT increased insignificantly in the presence of sMyBP-C aggregates compared to the monodisperse protein (Figure 6c). Revealed insignificant differences in ThT fluorescence can be caused by the absence of discrepancies in the secondary structure between monodisperse protein and sMyBP-C aggregates (Figure 6b). Insignificantly increased fluorescence of ThT in the presence of sMyBP-C aggregates is, possibly, due to the predominance of amorphous aggregates, but not fibrils, of this protein. Owing to an insignificant binding of ThT to sMyBP-C aggregates, they are amyloid-like rather than amyloid aggregates.

### 2.8. Small Angle X-ray Scattering of sMyBP-C 

SAXS was used to obtain information about the conformation of the molecular and aggregated forms of sMyBP-C directly in the solution. First, we assessed the molecular weights of its molecular and aggregated forms by comparison with the known weight of BSA at the extrapolation of the scattering curve to the zero scattering angle (see Figure 7b). The molecular weight of molecular sMyBP-C was 150 ± 10 kDa; of its aggregated form, 900 ± 60 kDa. The obtained values for the molecular form of sMyBP-C (150 ± 10 kDa) are very close to its true molecular weight (125–134 kDa), as well as to the values obtained at the assessment of its electrophoretic mobility (~125 kDa). As for the aggregated form, the value of 900 kDa appears to correspond to an oligomer consisting of 6–7 molecules of this protein. As is known from [43], plotting the scattering curve in the log *I*–log *Q* coordinates (where *I* is intensity; *Q*, the module of the scattering vector) one can roughly assess the conformation of a molecule from the tangent of the regression line (tan*A*) (tan*A* equals 1 for rod-like; 2, plate-like; 4, globular particles). Figure 7a shows that the log *I*–log *Q* dependence for the molecular form of sMyBP-C is linear and tan*A* = −1.1. This means that the molecular form of the protein is elongated (rod-like). We can assess the cross-section radius of gyration (*R*_c_) of this rod-like structure by plotting the dependence log(*IQ*) − *Q*^2^ [43]. In the case of a circular cylinder, *R*_c_ = *R*/2, where *R* is the radius of the cylinder. In Figure 7b, this dependence is given for the molecular form of sMyBP-C; the calculated *R*_c_ is 30.0 ± 2 Å. Additional information can be obtained by plotting the scattering curve in *IQ*^2^ vs. *Q* coordinates (Kratky plot).

Figure 7c is a Kratky plot for the molecular form of sMyBP-C. The dependence is bell-shaped, which implies a compact structure of the protein. From its maximum, we can assess the protein’s radius of gyration (*R*_g_) to be 30.0 ± 2 Å.

The same calculations as for the molecular form were made for the aggregated form of sMyBP-C. Figure 7d shows that the log *I*–log *Q* dependence for the aggregated form is not linear, the slope of the straight line is tan*A* = −1.7. This is indicative of a rod-plate conformation. Nevertheless, the log (*IQ*) vs. *Q*^2^ dependence (Figure 7e) is linear and the assessed *R*_c_ = 68 ± 2 Å.

The radius of gyration for aggregated sMyBP-C was assessed using a Kratky plot (Figure 7f). The calculated *R*_g_ was 86 ± 4 Å. 

The protein conformation can be assessed from SAXS data using the DAMMIF program [43]. Figure 8 presents assessed conformations for the molecular form of sMyBP-C (left) and the aggregated form of the protein (right). The aggregated forms are morphologically similar to sMyBP-C oligomers visualized using TEM and AFM (Figure 4 and Figure 5). The conformation of the protein molecule shown in Figure 8 on the left resembles the morphology of cardiac C-protein molecules visualized by electron microscopy and described in [44]. 

### 2.9. Cytotoxicity Study of sMyBP-C Aggregates and Oligomers

To elucidate whether amyloid-like aggregates of sMyBP-C have a negative effect on cells of the organism, we studied their toxicity on a smooth-muscle cell culture. Figure 9 shows the action of sMyBP-C aggregates on rat aortic smooth muscle cells. As can be seen in the plot, at a combined incubation for 72 h sMyBP-C oligomers and aggregates had no direct toxic effect on rat aortic smooth muscle cells within this concentration range. 

### 2.10. Calculation of Amino Acid Sequence Identity

The amino acid sequence identity between the domains was calculated for two paralogs of human sMyBP-C (ssMyBP-C and fsMyBP-C). The results are given in Table 1. As seen from the table, the mean identity between adjacent domains in both paralogs does not exceed 26%, which is relatively low. The maximum identity of 30% was registered between domains Fn1 and Fn2 for ssMyBP-C (Appendix A). 

### 2.11. Calculation of Disordered Regions in sMyBP-C Molecule 

According to CD data, the α-helix and β-sheet contents in the sMyBP-C molecule do not exceed 40% (Figure 6b). We elucidated the arrangement of disordered regions in the protein molecule. To reveal disorder, we analyzed the sMyBP-C amino acid sequence using the IsUnstruct program [45,46]. Figure 10 shows a schematic of sMyBP-C domain structure and probability profiles of disordered segments in the protein. The predicted disordered regions are shown in brown and have probabilities higher than 0.5. In general, the results for ssMyBP-C and fsMyBP-C are similar. Most disordered segments are in PA- and M-domains, which are closer to the N-terminus of the molecule. Other disordered segments are in C5 Ig-like and C6–C7 FnIII domains of ssMyBP-C and C5 Ig-like and C6 FnIII domains of fsMyBP-C. This region with many disordered segments is in the sequence 550–770 approximately in the mid part of the molecule. 

## 3. Discussion 

This work showed the ability of sMyBP-C isolated from rabbit skeletal muscles to form aggregates at a decrease of ionic strength. According to the data of DLS (Figure 3a) and TEM (Figure 4a,b), maximally large aggregates emerged already after 5–10 min, which is indicative of a rather high aggregation rate. TEM and AFM showed sMyBP-C to form large amorphous aggregates of ~2 up to ~10 µm in size, fibrils ~20 nm wide, and oligomers ~50–60 nm. TEM and DLS found the presence of sMyBP-C oligomers both at the initial stages of aggregation (5–10 min) and after 16–24 h. According to the data of SAXS and DLS, the sMyBP-C oligomer consists of 7–10 monomers. 

It should be noted that the occurrence of sMyBP-C oligomers at the initial stages of aggregation can be the cause of a high aggregation rate. As is known, the presence of oligomers (preformed seeds) in a preparation can significantly accelerate aggregation. In this case, the aggregation process begins with the elongation phase/growth phase, avoiding the nucleation phase/lag phase, as it has already been shown in other works for Aβ peptide [47]. 

X-ray diffraction analysis of sMyBP-C aggregates revealed two ring-shaped diffuse reflections at 4.6 and 10 Å (Figure 6a), corresponding to the quaternary cross-β structure characteristic of all amyloid fibrils. Diffuse rings on the diffraction pattern have been explained and described [48] in studies of non-oriented and weakly oriented amyloid samples. Similar diffuse ring reflections have been shown for amyloid aggregates of many proteins [10,40,49,50,51,52,53,54,55,56,57,58,59]. 

However, other data we obtained, such as the poor binding of ThT to sMyBP-C aggregates, the absence of changes in the secondary structure of the protein during aggregation, are not quite consistent with modern views of the amyloid aggregation process [60,61]. Nevertheless, X-ray diffraction data are indicative of the amyloid nature of sMyBP-C aggregates. No explanation of this contradiction was found. When discussing this issue, the features of the subject of research should be taken into account: sMyBP-C is a fibrillar protein of molecular weight ~130 kDa consisting of 10 Ig- and FnIII-like domains of a beta-folded structure. Very few studies have dealt with amyloid aggregation of proteins with such large molecular weights. Earlier [35,37], we have described the amyloid properties of aggregates of smooth-muscle titin, also consisting of Ig- and FnIII-like domains. As MyBP-C, titin formed amorphous aggregates with a cross-β quaternary structure without changes in the secondary structure. We assume that the formation of amyloid-like aggregates of MyBP-C and titin without changes in their secondary structure can occur due to a change of the tertiary structure as a consequence of a partial unfolding of domains free of S–S bonds, followed by the interaction of open β-strands between adjacent domains and/or domains of the adjacent molecules. This is quite possible considering the data that the unfolding of Ig/FnIII domains of sMyBP-C occurs at relatively low values of strength (∼30 to ∼150 pN) [62]. A similar partial unfolding of Ig/FnIII domains can also occur in titin [63,64,65]. It has been shown that a partial unfolding of particular Ig-like domains in the titin I-zone occurs at a low value of strength (about 10 pN) [66,67]. This indicates that a minimal amount of energy is used for the unfolding of titin domains. As molecules of sMyBP-C and titin consist of the same types of domains, we can assume a partial unfolding of sMyBP-C domains during the formation of a cross-β quaternary structure in the process of aggregation. Thus, considering the results we obtained, sMyBP-C can be said to form a special type of amyloid-like aggregates. 

There are data, which show that the aptitude for aggregation in multidomain proteins depends on the identity of the amino acid sequence between their domains. It is considered that the low aptitude for aggregation is characteristic of those proteins in which the identity between domains is less than 40% [68,69]. Individual domains of titin have been used as a model [68,69]. The authors of those works concluded that this feature formed most likely as a result of evolutionary pressure and was meant to avoid misfolding and subsequent aggregation. We calculated the identity of amino acid sequences for sMyBP-C (Table 1). The identity of the sMyBP-C amino acid sequence was found to be low. In particular, the maximal mean identity in the amino acid sequence for sMyBP-C was 26% (Table 1). These data, as for the case of titin, indicate a rather low aptitude of sMyBP-C for aggregation as laid down by nature on the level of the primary structure. It is, probably, for this reason that MyBP-C aggregates only at low ionic strength values and only in vitro, and that is why no aggregates of sMyBP-C have been found in vivo. 

Thus, the results of our study showed that myosin binding protein-C from rabbit skeletal muscle formed in vitro amyloid-like aggregates, which exhibited no toxicity on cell culture. The formation of a quaternary cross-β structure occurred without changes in the secondary structure of sMyBP-C. We presume that the quaternary cross-β structure can be formed in sMyBP-C aggregates due to a partial unfolding of its domains free of S–S bonds, followed by the interaction of open β-strands between adjacent domains and/or domains of adjacent molecules. Formation of amyloid or amyloid-like sMyBP-C aggregates in vivo is little probable due to a low identity in the amino acid sequence, the occurrence of alternating ordered and disordered regions in the protein molecule, as well as S–S bonds providing for the general stability of its molecules. The obtained data expand the views of the general mechanisms of protein aggregation. 

## 4. Materials and Methods

### 4.1. Purification of sMyBP-C from Rabbit Skeletal Muscles 

sMyBP-C isolated from rabbit skeletal muscles was used. All animal procedures were approved by the Commission on Biosafety and Bioethics (Institute of Theoretical and Experimental Biophysics, Russian Academy of Sciences; Permission No. 30 dated 10 September 2019) in accordance with Directive 2010/63/EU of the European Parliament. Rabbit surgeries were performed under anesthesia with Zoletil (Virbac Sante Animale, Carros, France) (4 mg/kg, i.m.); all efforts were made to minimize animal suffering. 

sMyBP-C was prepared from rabbit body and hindlimb skeletal muscles by the method described in [70]. The muscles were homogenized; myosin was extracted with a Guba–Straub solution (threefold volume with respect to muscle weight) containing 0.3 M KCl, 0.15 M potassium phosphate buffer, 1 mM PMSF, 20 μg/mL trypsin inhibitor, pH 6.5. Extraction was performed at 4 °C for 10–15 min with constant stirring, then the mixture was centrifuged for 15 min at 5000× *g*. The supernatant was resuspended with a 14-fold volume of cooled bidistilled water containing 0.1 mM DTT and 0.1 mM NaN_3_. After 40–60 min, the myosin residue was collected by centrifugation for 30 min at 2500× *g*. A solution containing 2 M KCl, 0.2 M potassium phosphate buffer, 4 mM EDTA, pH 7.0, to a final concentration of 0.5 M KCl, 0.05 M potassium phosphate buffer, 1 mM EDTA, pH 7.0, was added to the residue. After its dissolution, the residue was diluted with a solution containing 0.5 M KCl, 0.05 M potassium phosphate buffer, 1 mM EDTA, 0.1 mM NaN_3_, pH 7.0, to a protein concentration of 12 mg/mL for the subsequent fractionation with ammonium sulfate. 

An equal volume of 2.8 M ammonium sulfate cooled to 4 °C, in a solution containing 0.5 M KCl, 0.05 M potassium phosphate buffer, 1 mM EDTA, pH 7.0, was added to the protein solution (to a 36% saturation); the mixture was left for 20–30 min for an actomyosin residue to form, and then was centrifuged for 1 h at 2500× *g*. The residue was discarded; the supernatant was supplemented with a saturated (at 4 °C) ammonium sulfate solution up to 43% saturation. After 30 min, the formed residue that contained predominantly myosin and sMyBP-C was collected by centrifugation for 1 h at 2500× *g*. Then the residue was dissolved in a 0.15 M potassium phosphate buffer, pH 7.5, containing 10 mM EDTA, 0.1 mM DTT a 0.1 mM NaN_3_ (the column buffer), and dialyzed against the same buffer up to the complete removal of ammonium sulfate. 

Myosin clarified by centrifugation for 1 h at 10,000× *g* (pre-column myosin) contained about 5% sMyBP-C. To separate them, pre-column myosin was subjected to ion-exchange chromatography on a column with DEAE-Sephadex A-50 equilibrated with the column buffer. sMyBP-C was eluted in the free volume of the column and collected for further purification. Subsequently, sMyBP-C fractions obtained at the chromatographic purification of skeletal muscle myosin on a DEAE-Sephadex A-50 column were concentrated by ammonium sulfate to a 2.08 M saturation and sedimented by centrifugation for 1 h at 3000× *g*. The residue was dissolved in a buffer containing 0.3 M KCl, 4.8 mM K_2_HPO_4_, 5.2 mM KH_2_PO_4_, 0.1 mM DTT, 0.1 mM NaN_3_, pH 7.0, and dialyzed against the same buffer up to the complete removal of ammonium sulfate. The proteins were separated on a column of hydroxyapatite equilibrated in the same buffer. A phosphate gradient was used to remove the proteins from the column.

The concentration of sMyBP-C was determined spectrophotometrically by a Cary 100 spectrophotometer (Varian, Palo Alto, CA, USA), using the extinction coefficient (*E*_280_, 1 mg/mL) of 1.09 [71].

### 4.2. Gel Electrophoresis 

The purity of sMyBP-C was checked using SDS-PAGE by the technique of [72] with modifications. In our method, the separating gel contained 7% polyacrylamide instead of 15% (the acrylamide to *bis*-acrylamide ratio, 199:1), as well as 0.75 M Tris-HCl buffer, pH 8.8, 0.1% SDS, 10% glycerol, 0.05% tetramethylethylenediamine and 0.05% ammonium persulfate. Besides, instead of a stacking gel with the polyacrylamide content of 5% according to [72], we used a stacking gel as per [73] with 2.6–2.8% polyacrylamide (the acrylamide to *bis*-acrylamide ratio, 36.5:1). These modifications contributed to a better focusing of protein lanes in gel. The stacking gel also contained 0.125 M Tris-HCl buffer, pH 6.8, 0.1% SDS, 0.05% tetramethylethylenediamine and 0.05% ammonium persulfate. 

The electrode buffer in the electrophoresis contained 0.192 M glycine, 0.025 M Tris, and 0.1% SDS, pH 8.3. The electrophoresis was conducted at a current of 3–5 mA for the first 30–60 min, after that the current was increased to 12–15 mA. Upon the electrophoresis, the gels were fixed for 20–30 min in a solution containing 10% ethanol and 10% acetic acid. Then they were stained for 30–40 min in a solution containing 0.1% Coomassie G-250 and R-250 (mixed at a ratio of 1:1), 45% ethanol, and 10% acetic acid. The stained gels were washed in 7% acetic acid at constant stirring on a shaker. 

The molecular weight of sMyBP-C was assessed using TotalLab v1.11 software package (TotalLab, Newcastle Upon Tyne, UK). As molecular weight markers, we used a PageRuler Prestained Protein Ladder, 10 to 180 kDa (Thermo Fisher Scientific, Waltham, MA, USA). 

### 4.3. HPLC–MS Analysis of sMyBP-C 

Protein fractions were analyzed by tandem mass spectrometry. Proteins were treated with proteinase K (Promega, Madison, WI, USA) and trypsin (Sigma-Aldrich, St. Louis, MO USA). Then the peptide mixture was separated by reversed-phase chromatography (Easy nLC 1000; Thermo Fisher Scientific, Waltham, MA, USA) and analyzed by an OrbiTrap Elite ETD high-resolution mass spectrometer (Thermo Fisher Scientific, Waltham, MA, Germany). The potential difference between the emitter and the inlet cone was 1.8 kV; heated capillary temperature, 200 °C. Ion fragmentation was performed by collision activation in a high energy cell (HCD) and electron transfer dissociation (ETD). The masses of ions and ion fragments were recorded at a resolution of 60000 and 15000, respectively. The resulting fragmentation spectra were processed using the PEAKS Studio 7.5 software (Bioinformatics Solution Inc., Waterloo, ON, Canada). 

### 4.4. Conditions for the Formation of sMyBP-C Aggregates 

Purified sMyBP-C in buffer (0.3 M KCl, 4.8 mM K_2_HPO_4_, 5.2 mM KH_2_PO_4_, 0.1 mM DTT, 0.1 mM NaN_3_, pH 7.0) was used to form aggregates. sMyBP-C aggregates (concentration, 0.2–0.4 mg/mL) were formed by dialysis in cellulose membrane tubing (size, 25 × 16 mm^2^) (Sigma-Aldrich, St. Louis, MO, USA) for 1 and 24 h at 4 °C against solutions containing 0.15 M glycine-KOH, pH 7.0. 

### 4.5. Dynamic Light Scattering Experiments 

DLS experiments were conducted according to a protocol described in [35,37]. For DLS analysis of sMyBP-C aggregation, a protein sample in a buffer containing 0.3 M KCl, 4.8 mM K_2_HPO_4_, 5.2 mM KH_2_PO_4_, 0.1 mM DTT, 0.1 mM NaN_3_, pH 7.0, at an initial concentration of 1 mg/mL, was transferred into a solution of 0.15 M glycine–KOH, pH 7.0, by gradual dilution to a final concentration of 0.1 mg/mL to decrease ionic strength. Further steps were as in [37]. The collected autocorrelation functions were converted into particle-size distributions, using the general-purpose algorithm provided with the ZS Zetasizer Nano (Malvern Instruments Ltd., Malvern, UK) used in this experiment. Particle-size distributions obtained from alternative inversion algorithms yielded comparable results. Dynamic viscosity of the protein solutions determined using an SV-10 Sine-wave Vibro Viscometer (A&D Company Ltd., Tokyo, Japan) was 0.92 cP. This value was taken into consideration when measuring the particle dimensions in sMyBP-C samples collected 60 min after the dialysis. 

### 4.6. Transmission Electron Microscopy

A drop of aggregated protein suspension at a concentration of 0.1 mg/mL was applied to a carbon-coated collodion film (2% collodion solution in amyl acetate (Sigma-Aldrich, St. Louis, MO, USA)) on a copper grid (Sigma-Aldrich, St. Louis, MO, USA) and negatively stained with 2% aqueous uranyl acetate (SPI-Chem., West Chester, PA, USA). Samples were examined under a JEM-100B electron microscope (JEOL Ltd., Tokyo, Japan). 

### 4.7. Atomic Force Microscopy 

AFM was as described in [37]. AFM imaging was carried out using an AFM Ntegra-Vita microscope (NT-MDT Spectrum Instruments, Moscow, Russia) in noncontact (tapping) mode in air. The typical scan rate was 0.5–1 Hz. Measurements were carried out using NSG03 cantilevers with a resonance frequency of 47–150 kHz and ensured a 10 nm tip curvature radius. The processing and presentations of AFM images were performed using Nova PX software (NT-MDT Spectrum Instruments, Moscow, Russia) and Gwyddion 2.44 software (http://gwyddion.net/, Jihlava, Czech Republic). 

### 4.8. Fluorescence Analysis with Thioflavin T

The amyloid nature of sMyBP-C aggregates was estimated by the intensity of thioflavin T (ThT, Sigma-Aldrich, St. Louis, MO, USA) fluorescence (1 ThT: 5 sMyBP-C (*w*/*w*)). The fluorescence measurements are described in [35].

### 4.9. Circular Dichroism 

sMyBP-C was dialyzed for 24 h against the buffer containing 0.15 M glycine–KOH, pH 7.0–7.5. The CD spectra prior to and after sMyBP-C aggregation were recorded in a Jasco J-815 spectrometer (JASCO Inc., Tokyo, Japan) using 0.1 cm optical path-quartz cells and wavelengths of 250–190 nm. The secondary structure was calculated using the CONTIN/LL module of the CDPro program [74]. 

### 4.10. X-ray Diffraction 

X-ray diffraction analysis was carried out according to a protocol described in [35,37]. sMyBP-C aggregates for X-ray diffraction analysis were prepared after a 24-h incubation at 4 °C in a buffer containing 0.15 M glycine–KOH, pH 7.0–7.5. 

### 4.11. Small Angle X-ray Scattering 

SAXS was measured as in [75]. The distance between the sample and the source was 2.35 m. The range of detectable scattering vectors *Q* was 0.0080.2 Å^−1^ (*Q* = 4 π sinθ/λ, where λ is the wavelength of X-rays, 2θ is the scattering angle). The data were registered with a PILATUS 100 K two-dimensional CCD X-ray detector. The shape of particles was estimated from the slope (tanα) of log *I* dependence on log *Q*, where *I* is the intensity of scattering, and *Q* is the scattering vector [75]. 

### 4.12. Cytotoxicity Assay 

To obtain a high concentration of sMyBP-C, the protein was freeze-dried using a FreeZone 1 Liter Benchtop Freeze Dry System (Labconco, Kansas City, MO, USA). 1% trehalose was used as a stabilizing agent. The quality of sMyBP-C and any degradation after lyophilization were monitored in 7% SDS-PAGE [72]. To study the cytotoxicity of sMyBP-C aggregation, the protein in a buffer containing 0.3 M KCl, 4.8 mM K_2_HPO_4_, 5.2 mM KH_2_PO_4_, 0.1 mM DTT, 0.1 mM NaN_3_, pH 7.0, at an initial concentration of ~5 mg/mL was transferred to a solution of 0.15 M glycine–KOH, pH 7.0. The sample was diluted at 5 °C for 10 min for ionic strength to decrease gradually. Then, to separate oligomers and aggregates, the samples were centrifuged at 15,000 rpm for 10 min. It was assumed that the residue contained mainly aggregates and the supernatant, predominantly oligomers. The oligomers were additionally incubated for the next 24 h for their size to be maximum in the DLS assay. 

Cytotoxicity was assessed on smooth muscle cells isolated from rat aorta as described in [76] using the crystal violet assay [77]. The cells were seeded into 96-well cell culture plates (Greiner, Pleidelsheim, Germany) at a density of 3000 cells per well. Cells were cultured in a DMEM/F12 nutrient mixture (Sigma-Aldrich, St. Louis, MO, USA) with 10% FBS (Gibco, Gaithersburg, MD, USA), 40 µg/mL gentamycin sulfate (Sigma-Aldrich, St. Louis, MO, USA) to the confluent state, at 37 °C in an atmosphere containing 5% CO_2_. After the confluent formation, cells were incubated in DMEM/F12 without serum for 2 h. sMyBP-C was added in an oligomeric or aggregated form. Molecular sMyBP-C was used as a control. Cytotoxicity was estimated from the difference between the optical density in the experiment and the background to the difference between the control optical density and the background in 72 h incubation. Optical density was proportional to the number of living cells. Measurements were performed using an Infinite F200 plate reader (Tecan, Grödig, Austria). 

### 4.13. Calculation of the Identity of the Amino Acid Sequence and Disordered Regions in the sMyBP-C Molecule 

The sMyBP-C amino-acid sequence identity was calculated by the BLAST program. The data were retrieved from the UniProtKB databases: Q00872 (MYBPC1_HUMAN), Q14324 (MYBPC1_HUMAN), Q14896 (MYBPC1_HUMAN), G1TKC1 (MYBPC1_RABBIT), G1SGU7 (MYBPC3_RABBIT). The data for MyBP-C fast type were taken from the NCBI GenPept database: 655902480 (MYBPC2_RABBIT). 

### 4.14. Statistics

The results obtained in cytotoxicity experiments were statistically assessed using the Mann–Whitney *U* test with confidence levels *p* ≤ 0.05. The data presented are mean values (*M*) and standard errors (*m*). 

## Figures and Tables

**Figure 1 ijms-22-00731-f001:**
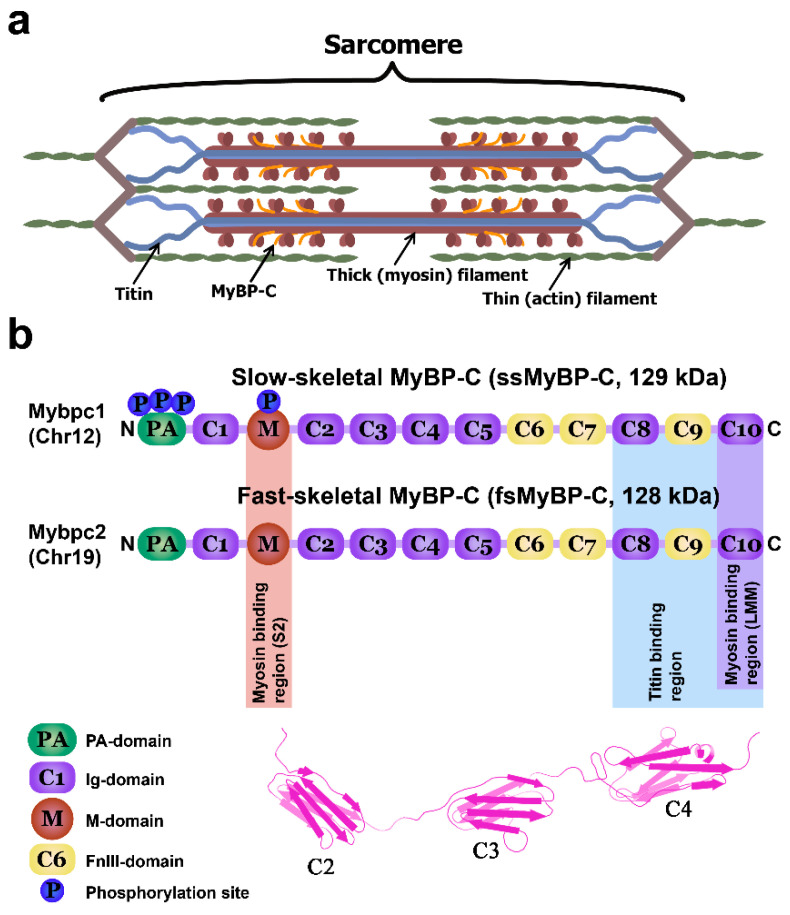
MyBP-C structure and location in the sarcomere. (**a**) Schematic structure of muscle sarcomere with thick and thin filaments. MyBP-C is localized in the C-zone and is shown in orange. (**b**) Domain structure of two paralogs of MyBP-C (based on schemes in [12,13]). Below, a schematic structure of three Ig-like ssMyBP-C domains (C2–C4). ssMyBP-C has three phosphorylation sites in the proline-alanine domain (PA-domain) and one within the M-domain. The region responsible for binding to titin (highlighted in blue) includes domains C8–C10. Interaction of domains C8–C10 with titin is shown for cardiac MyBP-C only [21,22,23]. Schematic diagram of structure C2–C3–C4 domains was drawn based on the PDB file https://www.uniprot.org/uniprot/Q00872.

**Figure 2 ijms-22-00731-f002:**
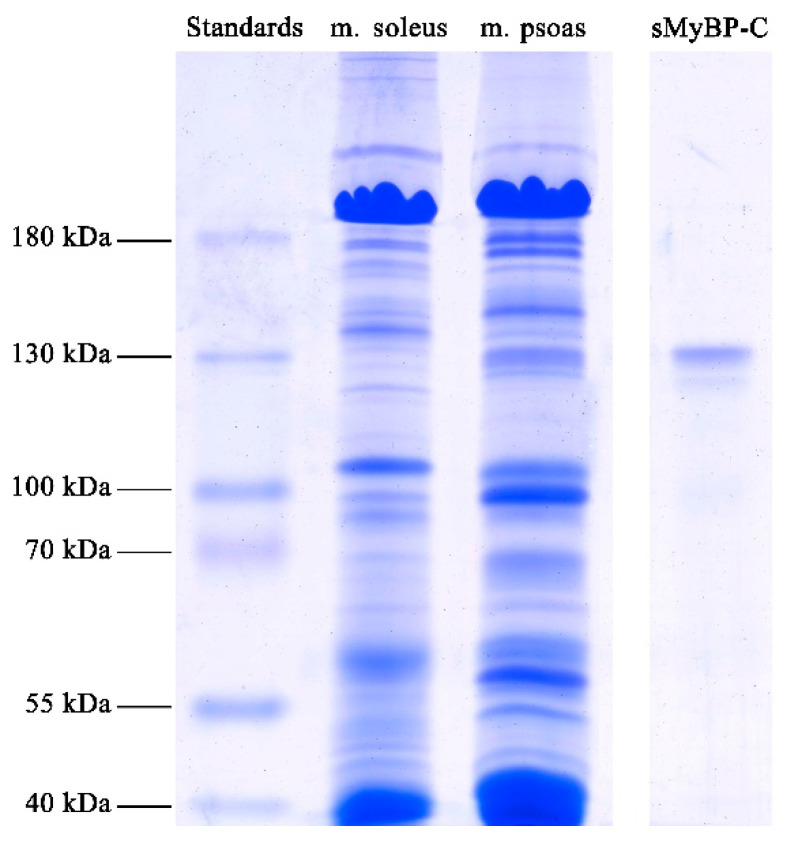
SDS-PAGE of purified sMyBP-C (right-hand lane). Rabbit skeletal muscles *m. soleus* and *m. psoas* were used as controls. Protein markers of molecular weights within the range of 15–180 kDa were also used (left-hand lane). The original images are at: https://drive.google.com/file/d/1THBQ23ezEtVeIx3_WKw6Py2Ql0ixHUVH/view?usp=sharing.

**Figure 3 ijms-22-00731-f003:**
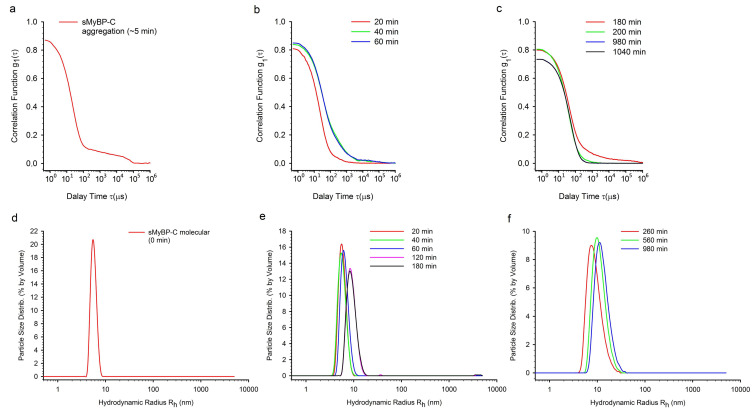
Kinetics of sMyBP-C aggregation monitored by DLS. (**a**–**c**) Evolution of field autocorrelation functions *g*_1_(*t*) of light scattered during sMyBP-C aggregate formation (pH 7.0–7.5, 10 °C). (**d**–**f**) Distribution of sMyBP-C particles. Formation of large aggregates and their time-dependent growth are shown.

**Figure 4 ijms-22-00731-f004:**
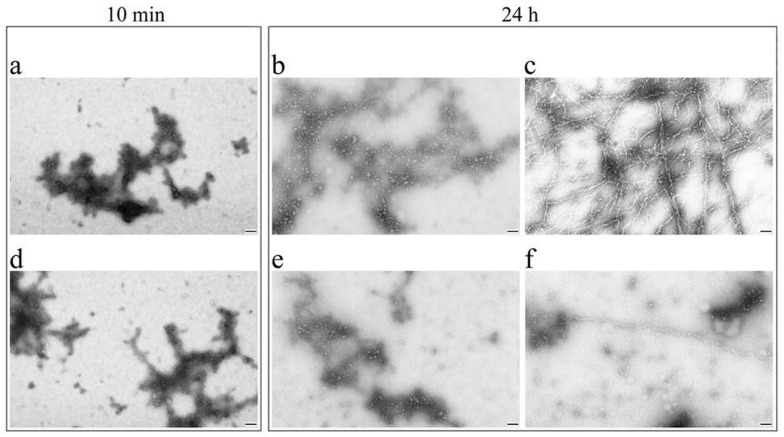
Electron microscopy of negatively stained sMyBP-C aggregates. (**a**,**d**) Amorphous aggregates and oligomers of sMyBP-C were obtained by placing it into a solution of 0.15 M glycine–KOH, pH 7.0, at 4 °C for 10 min; (**b**,**e**) sMyBP-C amorphous aggregates and oligomers as the main form of the aggregated protein were obtained by placing it into a solution of 0.15 M glycine–KOH, pH 7.0, at 4 °C for 24 h. (**c**) Fibrils of sMyBP-C were obtained by placing it into a solution of 0.15 M glycine–KOH, pH 7.0, at 4 °C for 24 h. (**f**) A separately lying fibrillar filament and oligomers of sMyBP-C were obtained by placing it into a solution of 0.15 M glycine–KOH, pH 7.0, at 4 °C for 24 h. Staining, by 2% aqueous uranyl acetate. Scale, 100 nm. The original images are at: https://drive.google.com/drive/folders/1UBx6m-Aa4Dl2In_PjiAweqmR6zyxVBz2?usp=sharing.

**Figure 5 ijms-22-00731-f005:**
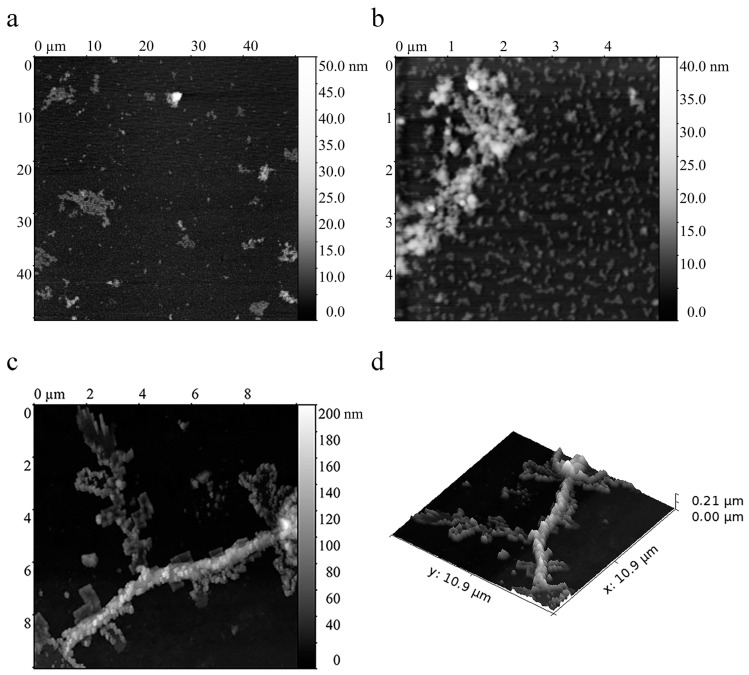
Atomic force microscopy of sMyBP-C aggregates formed after 24 h of incubation. (**a**,**b**) Large aggregates ~2 up to ~10 µm in size and up to 7–15 nm in height; (**b**) smaller aggregates shaped as extended structures up to 200 nm long ~50–60 nm wide and spherical structures ~50 nm in diameter. The height of these aggregates is from 10 up to 15 nm. (**c**,**d**) Large extended structures ~100 nm high and ~100 nm wide. The original images are at: https://drive.google.com/drive/folders/1oyOQtXYT475O34OxCol1TXhBkNg_k3fS?usp=sharing.

**Figure 6 ijms-22-00731-f006:**
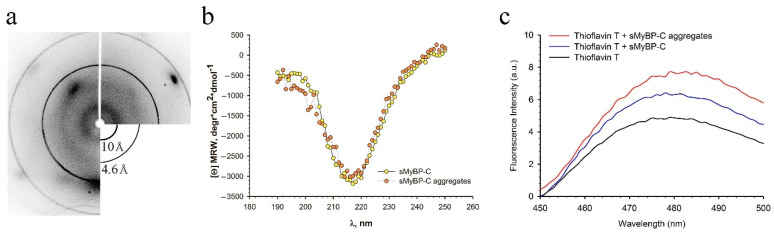
Corroboration of the amyloid properties of sMyBP-C aggregates by different methods. (**a**) X-ray diffraction of sMyBP-C aggregates after 24-h formation. Reflections were detected for sMyBP-C aggregates at 2.2, 3.1, 4.6, and 10 Å. The ~10 and ~4.6 Å diffuse reflections can be ascribed to a cross-β structure (Appendix A). The 2.2 and 3.1 Å reflections were not identified; (**b**) CD spectrum of sMyBP-C. The α-helix content of non-aggregated sMyBP-C (0.3 M KCl, 4.8 mM K_2_HPO_4_, 5.2 mM KH_2_PO_4_, 0.1 mM DTT, 0.1 mM NaN_3_, pH 7.0) was 4.5%; the β-structure was 30.7%. The helix content of aggregated sMyBP-C (0.15 M glycine–KOH, pH 7.0) was 4.9%; the β-structure, 29.7%; (**c**) Thioflavin T fluorescence in the presence of sMyBP-C aggregates (24-h formation). Black line, thioflavin T fluorescence; blue line, ThT fluorescence in the presence of non-aggregated sMyBP-C; red line, ThT fluorescence in the presence of sMyBP-C aggregates.

**Figure 7 ijms-22-00731-f007:**
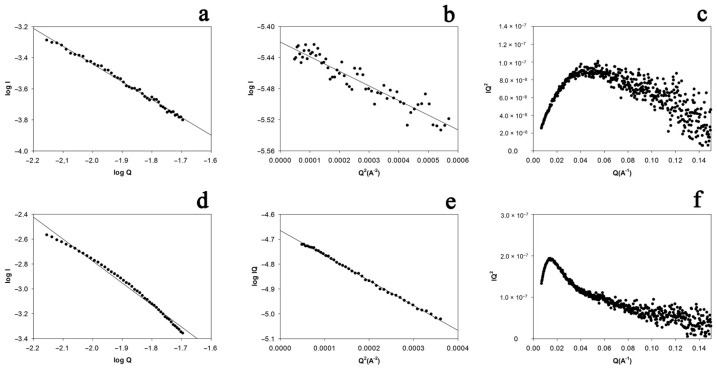
SAXS analysis of the molecular (**a**–**c**) and aggregated (**d**–**f**) forms of sMyBP-C.

**Figure 8 ijms-22-00731-f008:**
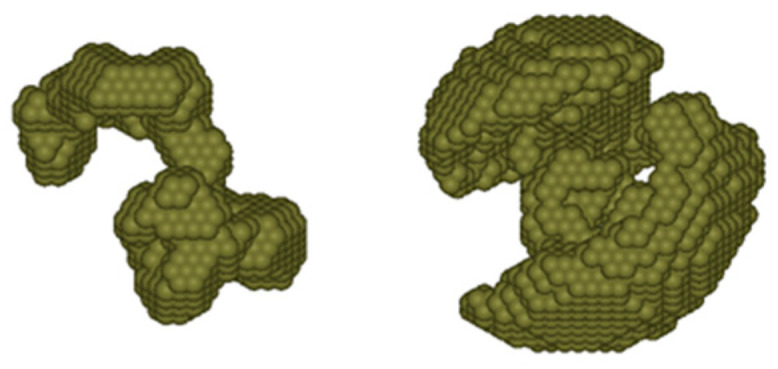
Possible conformations of the molecular (**left**) and aggregated (**right**) forms of sMyBP-C. SAXS data were assessed using the DAMMIF program [43].

**Figure 9 ijms-22-00731-f009:**
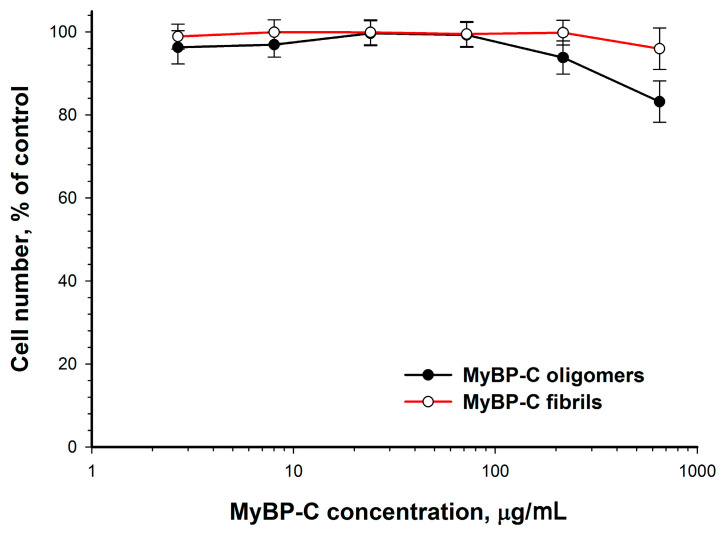
Effect of sMyBP-C aggregates (red graph) and sMyBP-C oligomers (black graph) on rat aortic smooth muscle cells after 72-h incubation.

**Figure 10 ijms-22-00731-f010:**
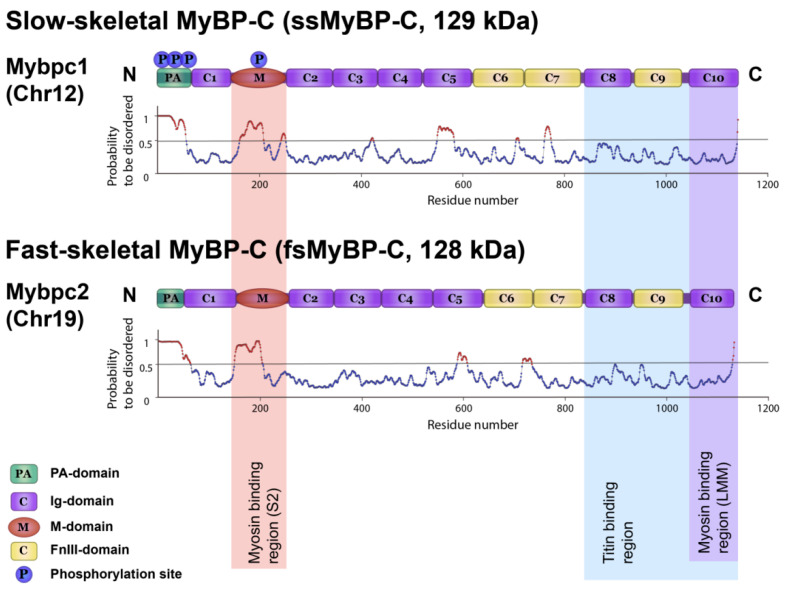
sMyBP-C domain structure schematic aligned with the protein disorder prediction plots by the IsUnstruct program [46]. Probabilities of ≥0.5 mean a disorder. Upper, ssMyBP-C paralog; lower, fsMyBP-C paralog.

**Table 1 ijms-22-00731-t001:** Identity of the amino acid sequence

Protein	Domain 1 (for Pair Comparison)	Number of Domains in the Group	Average Length of Domain	Domain 2 (for Pair Comparison)	(*Id*) Mean Identity, %	SID Standard Deviation/Dispersion of Identity, %	Number of Domain Pairs with Zero Identity, *N*_0_	Total Number of Domain Pairs, *N*	*N*_0_/*N* × 100, %
**ssMyBP-C**	Ig-like	7	90.1	Ig-like	13	10%	10	42	24
	Ig-like	7	90.1	FnIII-like	3	4%	11	21	52
	Ig-like	7	90.1	none	1	3%	10	14	71
	FnIII-like	3	102.7	FnIII-like	25	4%	0	6	0
	FnIII-like	3	102.7	none	4	6%	3	6	50
	none	2	88.5	none	0	0%	2	2	100
**fsMyBP-C**	Ig-like	7	96.7	Ig-like	14	9%	8	42	19
	Ig-like	7	96.7	FnIII-like	2	4%	14	21	67
	Ig-like	7	96.7	none	3	5%	8	14	57
	FnIII-like	3	96.3	FnIII-like	26	1%	0	6	0
	FnIII-like	3	96.3	none	3	3%	3	6	50
	none	2	75	none	8	0%	0	2	0

Identity was calculated by the formula: *Id* = 2 *N*_id_/(*L*_1_ + *L*_2_); *N*_id_, number of identical residues in an alignment; *L*, number of residues in a domain; ssMyBP-C domains were taken from http://www.uniprot.org/uniprot/Q00872; fsMyBP-C domains were taken from http://www.uniprot.org/uniprot/Q14324; For cases of over 50 residues between domains, a none pseudodomain was formed; FnIII-like, Fibronectin type-III.

## Data Availability

All data generated or analysed in the course of this research (including files of additional information) were incorporated into the article and Appendix A.

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
