# Peer review of "Myosin Binding Protein-C Forms Amyloid-Like Aggregates In Vitro"

_ijms, 2021, doi:10.3390/ijms22020731_

Round 1
Reviewer 1 Report
The study under the title “Myosin binding protein-C forms amyloid-like aggregates in vitro”, by Bobyleva et al, describes the aggregation related features of sMyBP-C in vitro under very specific condition - 0.15 M glycine–KOH, pH 7.0-7.5. They employed an extensive set of appropriate methodology to characterize the aggregation (kinetics) of sMyBP-C. They revealed the mixture of amorphous aggregates together with oligomeric species. They also provided some evidence for the appearance of cross beta (amyloid) structure (X-ray diffraction pattern).
However, I find the conclusions not supported enough by the experimental results in this study. Namely, the whole discussion part after the line 342 is very speculative in my opinion, and not strongly enough supported by the evidence. The emphasis of the paper is put on the explanation of the in vivo features of this protein and on the biological role of the aggregation – none of which is experimentally justified. E.g. what happens in vivo is discussed based for the most part on the amino acid composition comparison that authors provide, as well as on the prediction of the disordered regions. Furthermore, the aggregation detected under very specific conditions in vitro, is ascribed to the physiological role of the sMyBP-C in the sarcomere, without any experiments that confirmed what really happens in vivo or ex vivo.
Author Response
Comment
…I find the conclusions not supported enough by the experimental results in this study. Namely, the whole discussion part after the line 342 is very speculative in my opinion, and not strongly enough supported by the evidence. The emphasis of the paper is put on the explanation of the in vivo features of this protein and on the biological role of the aggregation – none of which is experimentally justified. E.g. what happens in vivo is discussed based for the most part on the amino acid composition comparison that authors provide, as well as on the prediction of the disordered regions. Furthermore, the aggregation detected under very specific conditions in vitro, is ascribed to the physiological role of the sMyBP-C in the sarcomere, without any experiments that confirmed what really happens in vivo or ex vivo.
Response
We took heed of Reviewer’s opinion and considerably abridged this part of the Discussion (after line 342). Whether sMyBP-C aggregation in vivo is possible, and whether it is of any functional significance for the cell are issues for future research. Thanks.
Also, the translator of our manuscript made minor stylistic changes to eliminate cases of tautology and redundancy. We hope that the manuscript reads better now.
Reviewer 2 Report
The manuscript submitted by Dr. Vikhlyantsev and Dr. Bobylev describes the formation of amyloid aggregates of Myosin binding protein-C (MyBP-C) in vitro. As a model, the skeletal myosin binding protein-C was selected. An impressive number of experimental techniques were used to characterize the structural nature of this aggregate. I confess that I lack competences in some of these biophysical methods (Dynamic light scattering and Atomic force microscopy in particular).
Major comments.
The principal limitation of this study is the fact that MyBP-C aggregates in vitro but not in vivo. It is therefore possible to ask why this reaction was studied. The Authors answer quite well to this question, in the Discussion, where they describe why sequence and structure themselves (and their evolution) may explain the absence of aggregation in vivo.
Apparently, secondary structure is unchanged during protein aggregation. This is rather surprising and perhaps the Authors might try to use another method for detecting the presence of strands: Fourier Transform Infrared Spectroscopy. It seems to be preferable, especially for beta-rich proteins.
Minor comments.
Lines 50-51. The Authors write that protein aggregation can be amyloid and non-amyloid but do not provide any reference and ignore the non-amyloid aggregation in the rest of the chapter. Perhaps, a short sentence on non-amyloid aggregation might be desirable.
Lines 96-97. Although the sentence “We believe that, due to its ability to interact in vivo with various proteins, this protein is apt to aggregation” is reasonable, it might need an explanation or a reference.
Lines 213-214. In the caption of Figure 6 the Authors write “Reflections were detected for sMyBP-C aggregates at 2.2, 3.1, 4.6 and 10 Å.” However, only the last two numbers (4.6 and 10) are shown in the image (Fig 6a). I would include also the first two (2.2 and 3.1). Moreover, I would comment them in the text.
Line 305. The expression “large aggregates emerged already after 5–10 min of aggregation” might become “large aggregates emerged already after 5–10 min”.
Author Response
Major comments
The principal limitation of this study is the fact that MyBP-C aggregates in vitro but not in vivo. It is therefore possible to ask why this reaction was studied. The Authors answer quite well to this question, in the Discussion, where they describe why sequence and structure themselves (and their evolution) may explain the absence of aggregation in vivo.
Apparently, secondary structure is unchanged during protein aggregation. This is rather surprising and perhaps the Authors might try to use another method for detecting the presence of strands: Fourier Transform Infrared Spectroscopy. It seems to be preferable, especially for beta-rich proteins.
Response
We consider MyBP-C as a model object for studying the features of protein aggregation on the whole. Our research is aimed to add to the understanding of the general problem of protein misfolding and, partially, to answer the question why some proteins form amyloid aggregates in the organism and others do not. We showed the ability of MyBP-C to form amyloid aggregates in vitro. Having proved by various methods that amyloid aggregation of MyBP-C is possible in vitro and what should be paid attention to, even under milder conditions than in other amyloid proteins and peptides (e.g., insulin and amyloid beta peptide), we showed its potential aptitude to this type of harmful aggregation. Then, having analyzed the obtained results and the literature data, we tried to explain within the framework of the Discussion why no amyloid-type aggregation of MyBP-C occurs in the organism.
We would also like to add that the problem of aggregation studies is very complex from the point of view of a practically absent methods of studying amyloid aggregation in vivo. Even the aggregation of amyloid beta peptide, deposits of which are found in Alzheimer’s disease, is studied under in vitro conditions. That said, the methods using which in vitro aggregation of proteins is investigated have strong limitations. These limitations are associated mainly with the size of amorphous aggregates and mature fibrils (which can reach tens of micrometres). For this reason, a maximum number of possible available methods should be used, which is what we tried to do.
As for the second part of the comment (with respect to Fourier transform infrared spectroscopy), it was rather difficult for us to obtain protein in sufficient amounts to prepare samples for a qualitative analysis by the FTIR method with an accuracy of secondary-structure decomposition exceeding that of the CD method. Samples of a concentration from 20 mg/ml and higher are required to obtain reliable data of the secondary structure using FTIR; several refills of the cuvette are required. In our preparations, the maximal concentration of MyBP-C was 5–6 mg/ml (CD assays require far lower concentrations, up to 1 mg/ml). At higher concentrations of MyBP-C, it stops to dissolve and the preparation turns into a gel.
Still, we should emphasize that if the Reviewer believes this experiment to be of crucial significance, we would make our best to produce the required amount / concentration of the protein to carry out the measurements.
Lines 50-51. The Authors write that protein aggregation can be amyloid and non-amyloid but do not provide any reference and ignore the non-amyloid aggregation in the rest of the chapter. Perhaps, a short sentence on non-amyloid aggregation might be desirable.
Response
We removed this sentence as our research deals with amyloid aggregation of proteins.
Comment
Lines 96-97. Although the sentence “We believe that, due to its ability to interact in vivo with various proteins, this protein is apt to aggregation” is reasonable, it might need an explanation or a reference.
Response
References were added. Thanks.
Comment
Lines 213-214. In the caption of Figure 6 the Authors write “Reflections were detected for sMyBP-C aggregates at 2.2, 3.1, 4.6 and 10 Å.” However, only the last two numbers (4.6 and 10) are shown in the image (Fig 6a). I would include also the first two (2.2 and 3.1). Moreover, I would comment them in the text.
Response
In the caption to Fig. 6, a phrase was added: The 2.2 and 3.1 Å reflections were not identified. In the text, we added a sentence about a possible origin of these reflections.
Comment
Line 305. The expression “large aggregates emerged already after 5–10 min of aggregation” might become “large aggregates emerged already after 5–10 min”.
Response:
The words of aggregation were removed. Thanks.
Round 2
Reviewer 1 Report
I find the new version of the Manuscript improved and my comments are satisfactorily addressed.